# *pruR* and PA0065 Genes Are Responsible for Decreasing Antibiotic Tolerance by Autoinducer Analog-1 (AIA-1) in *Pseudomonas aeruginosa*

**DOI:** 10.3390/antibiotics11060773

**Published:** 2022-06-06

**Authors:** Muhammad Reza Pahlevi, Keiji Murakami, Yuka Hiroshima, Akikazu Murakami, Hideki Fujii

**Affiliations:** 1Department of Oral Microbiology, Institute of Biomedical Sciences, Tokushima University Graduate School, Tokushima 770-8504, Japan; muhammadrezapahlevi8@gmail.com (M.R.P.); murakamik@mw.kawasaki-m.ac.jp (K.M.); yuka.hiroshima@tokushima-u.ac.jp (Y.H.); akimu@tokushima-u.ac.jp (A.M.); 2Department of Clinical Nutrition, Faculty of Health Science and Technology, Kawasaki University of Medical Welfare, Kurashiki 701-0193, Japan; 3Department of Biology, Keio University School of Medicine, Yokohama 223-8521, Japan

**Keywords:** antibiotic tolerance, *Pseudomonas aeruginosa*, autoinducer analog-1, *pruR*, PA0066-65-64

## Abstract

*Pseudomonas aeruginosa* infection is considered a high-risk nosocomial infection and is very difficult to eradicate because of its tolerance to antibiotic treatment. A new compound, autoinducer analog-1 (AIA-1), has been demonstrated to reduce antibiotic tolerance, but its mechanisms remain unknown. This study aimed to investigate the mechanisms of AIA-1 in the antibiotic tolerance of *P. aeruginosa*. A transposon mutant library was constructed using miniTn5pro, and screening was performed to isolate high tolerant mutants upon exposure to biapenem and AIA-1. We constructed a deletion mutant and complementation strain of the genes detected in transposon insertion site determination, *pruR* and PA0066-65-64, and performed killing assays with antibiotics and AIA-1. Gene expression upon exposure to biapenem and AIA-1 and their relationship to stress response genes were analyzed. High antibiotic tolerance was observed in Tn5-*pruR* and Tn5-PA0065 transposon mutants and their deletion mutants, Δ*pruR* and ΔPA0066-65-64. Complemented strains of *pruR* and PA0066-65-64 with their respective deletion mutants exhibited suppressed antibiotic tolerance. It was determined that deletion of PA0066-65-64 increased *rpoS* expression, and PA0066-65-64 affects antibiotic tolerance via the *rpoS* pathway. Additionally, antibiotics and AIA-1 were found to inhibit *pruR* and PA0066-65-64. This study proposed that *pruR* and PA0066-65-64 are members of the antibiotic tolerance suppressors.

## 1. Introduction

Antibiotic resistance poses a significant threat to global public health. Antibiotics are one of the most powerful tools for fighting life-threatening infections, but resistance constantly undermines their efficacy [1,2]. Antibiotic resistance is defined as the ability of microorganisms to grow at high concentrations of antibiotics, marked by a substantial increase in the minimum inhibitory concentration (MIC) [3]. Furthermore, antibiotic resistance disturbs progress in healthcare, food production, and life expectancy [2]. 

Moreover, antibacterial chemotherapy often fails to eradicate pathogens despite being sensitive to antibiotics, which could be caused by antibiotic tolerance [4]. Antibiotic tolerance is the ability of bacteria to maintain viability while ceasing growth in the transient doses of lethal antibiotic exposure [5]. While resistant microorganisms are characterized by a significant increase in MIC, tolerant microorganisms exhibit no changes in MIC [3,6]. The relationship between antibiotic resistance and tolerance remains unclear.

Antibiotic-resistant strains can emerge from enhanced antibiotic tolerance by cyclic antibiotic treatments [7]. In the clinical setting, bacteria isolated from patients with chronic cystic fibrosis who underwent recurrent antibiotic exposure displayed rapid development of antibiotic tolerance, which subsequently became resistance [8]. *Pseudomonas aeruginosa* (*P. aeruginosa*) is classified as a serious threat to public health, causing approximately 32,600 cases of hospitalized patients, 2700 deaths, and 767 million dollars in healthcare costs in the United States in 2017 [2]. *P. aeruginosa* infection is considered a high-risk nosocomial infection associated with high mortality, and is particularly dangerous for patients with chronic lung disease and immunosuppressed patients [2,9,10].

*P. aeruginosa* infection is difficult to eradicate because of its tolerance to antibiotics [4]. Several mechanisms, such as the two-component system, carbon metabolism, iron regulation, and stress response, are involved in antibiotic tolerance of *P. aeruginosa* [11]. Other studies have shown that nutrient starvation [12], stringent response [13], and stress response [14] mediate tolerance formation in *P. aeruginosa*.

Quorum sensing (QS) is a bacterial cell-to-cell communication mechanism that regulates virulence gene expression in a cell density-dependent manner [15]. It involves the emission of *N*-acyl homoserine lactone, an autoinducer (AI), to activate a transcription factor that functions as an activator of gene expression [16,17]. An analog of this autoinducer, autoinducer analog-1 (AIA-1), has been demonstrated to reduce antibiotic tolerance and enhance the antibacterial activity of various antibiotics, such as biapenem, levofloxacin, and several macrolides, without affecting their MIC and without directly affecting QS [18,19]. However, the mechanisms underlying AIA-1 activity remain unclear. This study aimed to investigate the mechanisms by which AIA-1 decreases antibiotic tolerance, including the genes that are affected by this compound and their dynamics in the antibiotic tolerance mechanisms of *P. aeruginosa*.

## 2. Results

### 2.1. Two Transposon Mutants Exhibited High Antibiotic Tolerance

Transposon mutant library construction was successfully performed by mating wild-type *P. aeruginosa* (PAO1) with *Escherichia coli* harboring miniTn5pro. More than 3700 PAO1-Tn5 mutants were screened in the presence of biapenem and AIA-1. Two mutants, Tn5-*pruR* and Tn5-PA0065, with high antibiotic tolerance were selected for further study. High antibiotic tolerance was defined as a survival rate that was at least 10 times higher than that of PAO1 after exposure to a combination of antibiotics and AIA-1. These mutants were named based on their transposon insertion site determination. 

Tn5-*pruR* and Tn5-PA0065 had 316 and 967 times higher survival rates than those of PAO1, respectively, after exposure to biapenem and AIA-1. These results demonstrate that the disruption of *pruR* and PA0065 by Tn5 also disrupts the function of AIA-1 in reducing antibiotic tolerance, resulting in high antibiotic tolerance even after exposure to biapenem and AIA-1. The survival rates of PAO1 and the selected mutants are shown in Figure 1.

Transposon insertion site determination revealed that the Tn5-*pruR* mutant exhibited the Tn5 insertion in the *pruR* gene, which encodes the proline utilization regulator. Tn5-PA0065 exhibited the Tn5 insertion in the PA0065 gene, a gene member of the PA0066-65-64 operon. Each member of the operon encodes a hypothetical protein. The amino acid sequence of each member of this operon was uploaded to the protein BLAST website (blast.ncbi.nlm.nih.gov/Blast.cgi, accessed on 22 March 2022). We found that PA0066 encodes a gamma carbonic anhydrase family protein, PA0065 encodes a haloacid dehalogenase (HAD) family hydrolase, and PA0064 encodes a putative membrane protein.

### 2.2. Antibiotic Susceptibility of Transposon Mutants and Deletion Mutants

The MIC and minimum bactericidal concentration (MBC) of biapenem, levofloxacin, tobramycin, and AIA-1 to PAO1; the transposon mutants; and deletion mutants were measured. All tested antibiotics, as well as AIA-1, had comparable MIC and MBC to PAO1, the transposon mutants, and deletion mutants, as indicated in Table 1. These results demonstrated that there was no resistance formation upon the mutation of *pruR*, PA0065, or the PA0066-65-64 operon.

### 2.3. High Antibiotic Tolerance of Deletion Mutants

To elucidate the role of *pruR* and PA0066-65-64 in AIA-1 mechanisms and their relation to antibiotic tolerance, a killing assay with antibiotics, or a combination of antibiotics and AIA-1, was performed. Δ*pruR* and ΔPA0066-65-64 exhibited high antibiotic tolerance when exposed to antibiotics and AIA-1. The survival rates of Δ*pruR* and ΔPA0066-65-64 were 58 and 85 times higher than those of PAO1 (Figure 2A) after biapenem (32 mg/L) and AIA-1 (32 mg/L) exposure, respectively, and 111 and 160 times higher than those of PAO1 after exposure to levofloxacin (4 mg/L) and AIA-1 (Figure 2B), respectively. The survival rate of both deletion mutants was almost the same as those of PAO1 in the tobramycin killing assay (Figure 2C). 

### 2.4. Antibiotic Tolerance of the Complemented Strains

To further investigate the role of these genes in antibiotic tolerance, a complemented strain of each deletion mutant was constructed and a killing assay was performed with biapenem and levofloxacin. The Δ*pruR*/*pruR* and ΔPA0066-65-64/PA0066-65-64 complemented strains had 5800 times (Figure 3A) and 478 times (Figure 3B) lower survival rates than each of its vector controls, respectively, in the killing assay with biapenem and AIA-1. The survival rates of both complemented strains were comparable to those of the PAO1 vector control after exposure to biapenem and AIA-1.

The same pattern was observed in the levofloxacin killing assay, although a lower killing effect was observed in the Δ*pruR*/*pruR* complemented strain. The killing assay with a combination of levofloxacin and AIA-1 for the *pruR* complemented strain resulted in only a seven times lower survival rate than its vector control (Figure 3C), whereas the PA0066-65-64 complemented strain exhibited a 46 times lower survival rate than its vector control (Figure 3D). The survival rate of the ΔPA0066-65-64/PA0066-65-64 strain was comparable to those of the PAO1 vector control after exposure to levofloxacin and AIA-1. These results demonstrate that *pruR* and PA0066-65-64 play a role in suppressing antibiotic tolerance. 

### 2.5. Gene Expression of pruR and PA0066-65-64 Genes

Quantitative real-time PCR (qRT-PCR) was performed to investigate *pruR* and PA0066-65-64 expression in PAO1 upon exposure to biapenem (32 mg/L) and AIA-1 (32 mg/L). Interestingly, similar expression patterns were observed for *pruR* and PA0066-65-64. The expression of both genes was significantly lower in the biapenem group, as well as in the AIA-1 group compared to the control group. The combination of biapenem and AIA-1 restored the expression of both genes, although their expression levels were comparable to those of the control group (Figure 4A,B). These results show that biapenem and AIA-1 inhibited *pruR* and PA0066-65-64.

Further analysis was performed to elucidate the relationship between *pruR* and PA0066-65-64 with the *rpoS* gene. The *rpoS* gene is known as a regulator of *P. aeruginosa* antibiotic tolerance in the stationary phase and under heat stress [14]. qRT-PCR analysis revealed that after 12 h of culture, *rpoS* expression in ΔPA0066-65-64 was significantly higher than that in PAO1. *rpoS* expression in Δ*pruR* was comparable to that in PAO1 (Figure 4C).

## 3. Discussion

Analogs of *P. aeruginosa* autoinducers were first described by Smith et al. [20] A library of autoinducer analogs was generated by modifying homoserine lactone (HSL) in acyl-HSL with various amines and alcohols. Several autoinducer analogs have been described to have agonist and antagonist activities in the quorum-sensing system [21]. Another study also found agonist and antagonist effects of *N*-acyl *L*-homoserine lactone analogs in *P. aeruginosa* specific to *LasR* and *QscR* system receptors [22]. 

*N*-Acyl cyclopentylamides, another analog of autoinducers, has been demonstrated to inhibit QS in *P. aeruginosa* [23]. The autoinducer analog-1 (AIA-1) of *P. aeruginosa,* which was used in this study, had the ability to reduce antibiotic tolerance and enhance the antibacterial activity of antibiotics, but did not inhibit the QS system of *P. aeruginosa* [18,19]. However, the mechanisms by which AIA-1 affects antibiotic tolerance remain unclear.

In the present study, we identified *pruR* and PA0065 genes, which are closely associated with antibiotic tolerance that is affected by AIA-1, through PAO1-Tn5 transposon mutant screening. Tn5-*pruR* exhibits a transposon insertion site on the *pruR* gene, which encodes a proline utilization regulator. This gene was shown to regulate the virulence of the *P. aeruginosa* PAK strain under high concentrations of proline and glutamate [24]. Tn5-PA0065 has a transposon insertion site on the PA0065 gene, a member of the PA0066-65-64 operon. Amino acid sequence BLAST analysis revealed that the PA0066 amino acid sequence encodes gamma carbonic anhydrase (CA), PA0065 encodes haloacid dehalogenase (HAD) family hydrolase, and PA0064 encodes a putative membrane protein.

Gamma CA was characterized from *E. coli* and catalyzes the CO_2_ hydratase reaction to bicarbonate and protons [25]. This enzyme has also been characterized in another Pseudomonas species: *Pseudomonas fragi* [26]. The HAD family hydrolase has been characterized in *E. coli* as an enzyme that improves sorbitol production in recombinant/engineered cyanobacteria [27,28]. However, no studies have described PA0064 or its putative protein product.

These genes are potentially linked to AIA-1 mechanisms of antibiotic tolerance. This study is the first to analyze AIA-1 mechanisms in affecting antibiotic tolerance via these genes, and the first to elucidate the role of *pruR* and PA0066-65-64 in antibiotic tolerance. Deletion mutants of *pruR* and PA0066-65-64 exhibited high antibiotic tolerance to biapenem and levofloxacin, implying that both genes are linked to AIA-1 in reducing antibiotic tolerance. Exposure to ampicillin and ofloxacin has been demonstrated to induce overlapping changes in gene expression profiles [29]. This explains the pattern observed in our study, where biapenem and levofloxacin had similar effects on our strains.

The high antibiotic tolerance properties in the transposon and deletion mutants of *pruR* and PA0066-65-64 prompted us to propose these genes as members of the antibiotic tolerance suppressor family. Taniguchi et al. reported a gene, *tcp* (PA0561), that has a suppressive role in antibiotic tolerance. Deletion of *tcp* resulted in a significantly higher survival rate after carbapenem exposure compared to that of the parental strain [30]. 

To confirm this phenomenon, *pruR* and PA0066-65-64 genes were complemented to their respective deletion mutants. Complementation of *pruR* and PA0066-65-64 eliminated the high antibiotic tolerance property of their deletion mutants, reducing their survival rate to a level comparable to that of the PAO1 vector control. This event confirmed that *pruR* and PA0066-65-64 are antibiotic tolerance suppressors. However, in the *pruR* complemented strain, exposure to levofloxacin and AIA-1 only resulted in a seven times lower survival rate than that of the vector control. Thus, *pruR* may partially regulate levofloxacin tolerance in *P. aeruginosa*.

To determine the mechanisms involved in the interaction of *pruR* and PA0066-65-64 with AIA-1, qRT-PCR was performed to detect the expression of these genes in PAO1 under exposure to biapenem and AIA-1. Biapenem and AIA-1 significantly inhibited the expression of both genes in the wild-type *P. aeruginosa*. The combination of biapenem and AIA-1 restored the expression of both genes, although only to a level comparable to that of the control. Therefore, other genes may be involved in antibiotic tolerance suppression.

The *rpoS* gene has been described as the master regulator of antibiotic tolerance against biapenem, imipenem, ofloxacin, and other sources of stress such as heat and osmotic stress [14,31]. We previously reported that *rpoS* expression was clearly repressed by the addition of AIA-1, and that this repression partially affects antibiotic tolerance [18,19]. In this study, we focused on the relationship between *pruR* and PA0066-65-64 with the *rpoS* gene. Our results demonstrated that ΔPA0066-65-64 exhibited higher expression of *rpoS* than PAO1, whereas in Δ*pruR,* the expression was comparable to that of PAO1. The absence of PA0066-65-64 increased the expression of *rpoS* gene. Based on these findings, PA0066-65-64 may inhibit *rpoS* in decreasing antibiotic tolerance, albeit partially. In other words, antibiotic tolerance suppression factors may inhibit antibiotic tolerance by directly or partially inhibiting *rpoS*. Thus, inhibition of *rpoS* either directly by AIA-1 or via an antibiotic tolerance suppressor factor resulted in a higher killing effect of antibiotics.

In conclusion, our study proposed *pruR* and PA0066-65-64 as members of the antibiotic tolerance suppressor family. These genes were inhibited by antibiotics and AIA-1, whereas the combination of antibiotics and AIA-1 restored gene expression to a level comparable to that of the control, suggesting that there are more genes involved in antibiotic tolerance suppression. PA0066-65-64 had the function of partially inhibiting stress response genes. Further studies in the future should investigate the effects of antibiotics and AIA-1 on the gene products of *pruR* and PA0066-65-64.

## 4. Materials and Methods

### 4.1. Reagents

The autoinducer analog-1 used in this study was *N*-(piperidin-4-yl)-dodecanamide [18]. Biapenem was purchased from Meiji Seika Pharma Co. Ltd. (Tokyo, Japan). Gentamicin was purchased from the Wako Pure Chemical Corporation (Osaka, Japan). Levofloxacin was purchased from Sigma-Aldrich (St. Louis, MO, USA). Tobramycin was purchased from LKT Laboratories Inc. (St. Paul, MN, USA).

### 4.2. Bacterial Strains and Growth Condition

The bacterial strains and plasmids used in the present study are listed in Appendix A. Bacteria were grown in lysogeny broth (LB) medium (for *P. aeruginosa* PAO1 and deletion mutants), LB supplemented with 20 mg/L and 200 mg/L gentamicin (for *E. coli* and *P. aeruginosa* PAO1-Tn5 mutant, respectively), or LB supplemented with 0.2% arabinose (for complemented strains)

### 4.3. Primers

Primers used in this study are listed in Appendix A. Primers for bacterial strain construction were designed using the In-Fusion^®^ Primer Design tool (Web tool, https://www.takara-bio.co.jp/research/infusion_primer/infusion_primer_form.php, accessed on 4 June 2020) (Takara, Shiga, Japan) and primers for qRT-PCR were designed using the Primer Express^®^ 3.0 software (Applied Biosystems, Foster City, CA, USA).

### 4.4. Bacterial Strains Construction

#### 4.4.1. Transposon Mutants

Transposon mutagenesis was performed by mating wild-type *P. aeruginosa* (PAO1) with *E. coli* S17-1-λ *pir* harboring the pUT-miniTn5pro [32]. It consists of gentamicin-resistant genes and the R6K_γ_
*ori* region, which is a useful region for insertion site detection. Vogel-Bonner minimal medium plates, containing 200 mg/L gentamicin, were used for mutant selection. This mating produced transposon mutants referred to as PAO1-Tn5 mutants.

#### 4.4.2. Deletion Mutant

The deletion mutant strain was generated as previously described [18]. Briefly, 500 base pairs from the upstream and downstream region of the target gene were amplified from the PAO1 genomic DNA. An In-Fusion Cloning Kit (Takara, Shiga, Japan) was used to linearize both amplified fragments in a BamHI-EcoRI-digested pEX18Gm vector [33]. The infusion plasmid was transformed into *E. coli* XL2-Blue Ultracompetent cells (Agilent Technologies, Santa Clara, CA, USA) for plasmid propagation. A deletion mutant was obtained by mobilizing the infusion plasmid from *E. coli* S17-1 to PAO1 by mating. Deletion mutants were selected based on gentamicin susceptibility and loss of sucrose susceptibility. Construction of the deletion mutant was confirmed by sequencing.

#### 4.4.3. Complementation Strain

The gene target was amplified and aligned to the SpeI-HF-SacI-HF-digested pJN105 vector, which contains an arabinose-inducible promoter [34]. The aligned plasmids were transformed into *E. coli* NEB 5α, and the propagated plasmids were electroporated into *E. coli* S17-1. Subsequent mating with the deletion mutant produced a complemented strain. A vector control strain was also constructed by inserting an intact pJN105 plasmid into the deletion mutant and PAO1 through the aforementioned process.

### 4.5. Screening of Transposon Mutants

#### 4.5.1. Initial Screening

The PAO1-Tn5 mutants were picked from the selection plates and grown in 200 μL LB supplemented with 200 mg/L gentamicin inside multi-well tissue culture test plates (96-well plates, Zellkulturtestplate 96U, TPP, Trasadingen, Switzerland). Cells were grown to the stationary phase, achieved by overnight incubation at 37 °C with aeration. The samples were inoculated into 200 μL fresh liquid medium and then grown to the middle of the log phase (OD_595_ of 0.15–0.2). Biapenem and AIA-1 were added at a concentration of 32 mg/L. After 24 h of incubation, 10 µL of each sample was subcultured on LB agar plates. Mutants that exhibited significant colony growth after exposure to biapenem and AIA-1 were selected for the final screening.

#### 4.5.2. Final Screening

The same culture conditions were applied as in the initial screening, except that, in this step, the mutants were grown in 3 mL liquid medium inside a 15 mL conical centrifuge tubes (Thermo Fisher Scientific, Rochester, NY, USA). Biapenem or a combination of biapenem and AIA-1 at a concentration of 32 mg/L was added. Viable cell counts were determined before and after drug addition by subculturing 100 μL the sample on LB agar plates. By assuming that the colony-forming units (CFU) were 100% of the CFU value on plates subcultured before drug addition, mutant survival was quantified. Mutants that exhibited high antibiotic tolerance despite the presence of biapenem and AIA-1 were selected. High antibiotic tolerance was defined as a survival rate of up to 10 times or higher than that of PAO1 after exposure to a combination of biapenem and AIA-1.

### 4.6. Transposon Insertion Site Determination

The method for determining the site of pUT-miniTn5pro insertion has been previously described by Siehnel et al. [32]. The following modifications were applied in this study: (i) the KpnI restriction enzyme was used, (ii) *E. coli* JM109-*λ pir* competent cells were used for ligation mixture transformation, and (iii) LB agar supplemented with 20 mg/L gentamicin was used as the selection medium. Self-ligated plasmids from the colonies were isolated using the GenElute^TM^ Plasmid Miniprep Kit (Sigma Aldrich, St. Louis, MI, USA) and then sequenced using primers that annealed to the transposon and chromosome transposition junction. The Pseudomonas Genome Database (www.pseudomonas.com, accessed on 13 May 2020) [35] was used to determine the sequence location throughout the *P. aeruginosa* chromosomal sequence using BLAST sequence analysis.

### 4.7. Susceptibility Assay

The MIC and MBC of biapenem, levofloxacin, tobramycin, and AIA-1 were determined using the standard microbroth dilution method [36].

### 4.8. Killing Assay

The deletion mutant and PAO1 were grown in 3 mL LB medium inside a 15 mL conical centrifuge tubes to the mid-log phase (OD_595_ of 0.15–0.20). The following antibiotics and drugs were used in this assay: biapenem (32 mg/L), levofloxacin (4 mg/L), tobramycin (32 mg/L), and AIA-1 (32 mg/L). Survival rates were quantified 24 h (biapenem) or 4 h (levofloxacin and tobramycin) after drug addition. 

Killing assay on the complemented strains was performed with the same condition as aforementioned protocol. Complemented strains were grown in 3 mL LB supplemented with 0.2% arabinose. Biapenem (32 mg/L), levofloxacin (4 mg/L), and AIA-1 (32 mg/L) were used in this assay.

### 4.9. Quantitative RT-PCR

#### 4.9.1. *pruR* and PA0066-65-64 Expression Analysis

PAO1 was grown in 10 mL LB broth inside a 50 mL conical centrifuge tubes (Thermo Fisher Scientific, Rochester, NY, USA) until the late log phase (OD_595_ = 0.3). Biapenem, AIA-1, or a combination of both were added. The samples were collected before and 8 h after drug administration. Prior to RNA isolation, the samples were protected with the RNAprotect^®^ Bacteria Reagent (Qiagen, Tokyo, Japan). Total RNA isolation was performed under RNase-free conditions using a Nucleospin RNA isolation kit (Macherey-Nagel, Düren, Germany), and reverse transcription was performed using PrimeScript^TM^ RT reagent kit with gDNA Eraser (Perfect Real Time) (Takara, Shiga, Japan). A StepOnePlus^TM^ Real-Time PCR System with FAST SYBR^®^ Green Master Mix (Thermo Fisher Scientific, Vilnius, Lithuania) was used for qRT-PCR. *pruR* and PA0066-65-64 expression were analyzed using *rpsL* as the internal control.

#### 4.9.2. Stress Response Gene Expression Analysis

PAO1, Δ*pruR* and ΔPA0066-65-64 were grown under the same culture conditions described above. The samples were collected before and 12 h after inoculation. This procedure was performed without the addition of any drug. RNA isolation and qRT-PCR were performed as previously described, and *rpoS* expression in each strain was analyzed.

### 4.10. Statistical Analysis

All statistical analyses were performed using GraphPad Prism 9.3.1 (GraphPad Software, Inc., San Diego, CA, USA). Data are presented as mean and standard deviation. Data were analyzed using one-way ANOVA, and the *p* value was set at *p* < 0.05.

## Figures and Tables

**Figure 1 antibiotics-11-00773-f001:**
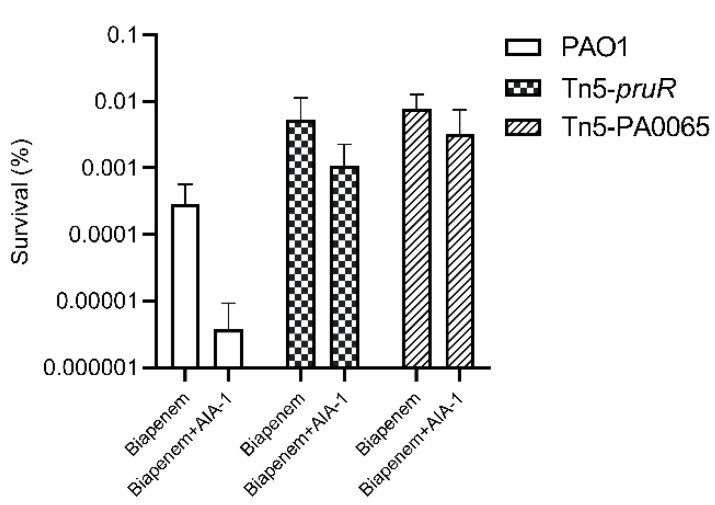
The survival rate of PAO1, Tn5-*pruR,* and Tn5-PA0065 in a killing assay using biapenem (32 mg/L) and a combination of biapenem and autoinducer analog-1 (AIA-1) (32 mg/L). Error bars show SD for three independent experiments.

**Figure 2 antibiotics-11-00773-f002:**
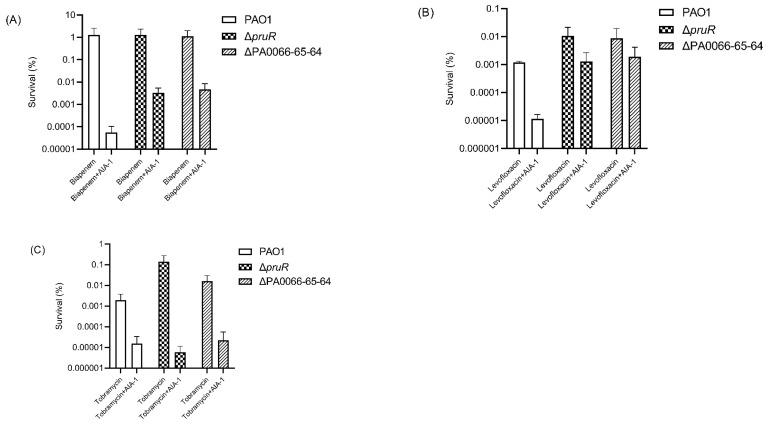
Killing effect of antibiotics or a combination of antibiotics and AIA-1 on PAO1 and deletion mutants. Survival rate of PAO1 and deletion mutants in a (**A**) biapenem killing assay, (**B**) levofloxacin killing assay, and (**C**) tobramycin killing assay. Error bars show SD for three independent experiments.

**Figure 3 antibiotics-11-00773-f003:**
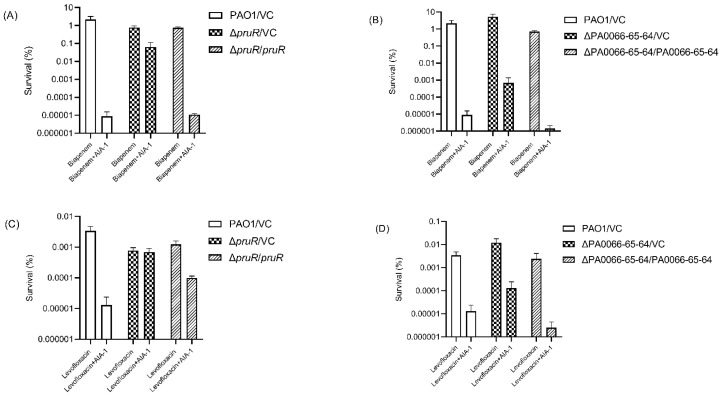
Killing assay with antibiotics and AIA-1 on complemented strains. Killing effect of biapenem and AIA-1 on (**A**) Δ*pruR*/*pruR*, Δ*pruR* vector control, and PAO1 vector control, and (**B**) ΔPA0066-65-64/PA0066-65-64, ΔPA0066-65-64 vector control, and PAO1. Killing effect of levofloxacin and AIA-1 on (**C**) Δ*pruR*/*pruR*, Δ*pruR* vector control, and PAO1 vector control, and (**D**) ΔPA0066-65-64/PA0066-65-64, ΔPA0066-65-64 vector control, and PAO1. Error bars show SD for three independent experiments.

**Figure 4 antibiotics-11-00773-f004:**
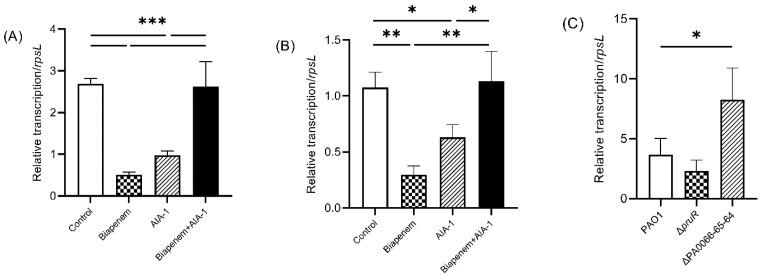
Expression of (**A**) *pruR* and (**B**) PA0066-65-64 in PAO1 after 8 h of exposure to biapenem, AIA-1, and combination of both, and (**C**) expression of *rpoS* in deletion mutants following 12-h culture, gene expression was normalized to *rpsL*, *n* = 3, * *p* < 0.05, ** *p* < 0.01, *** *p* < 0.001.

**Table 1 antibiotics-11-00773-t001:** Minimum inhibitory concentration (MIC) and minimum bactericidal concentration (MBC) of AIA-1 and various antibiotics to Δ*pruR* and ΔPA0066-65-64.

Strain	Biapenem		Levofloxacin		Tobramycin		AIA-1	
	MIC (mg/L)	MBC (mg/L)	MIC (mg/L)	MBC (mg/L)	MIC (mg/L)	MBC (mg/L)	MIC (mg/L)	MBC (mg/L)
PAO1	0.5	1	0.5	1	2	4	64	128
Tn5-*pruR*	0.5	1	0.5	1	4	8	64	64
Tn5-PA0065	0.5	2	0.5	1	4	8	64	64
Δ*pruR*	0.5	0.5	0.5	1	2	2	64	64
ΔPA0066-65-64	0.5	1	0.5	1	2	2	64	64

## Data Availability

The data presented in this study are available on request from the corresponding author.

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
