# Peer review of "pruR and PA0065 Genes Are Responsible for Decreasing Antibiotic Tolerance by Autoinducer Analog-1 (AIA-1) in Pseudomonas aeruginosa"

_antibiotics, 2022, doi:10.3390/antibiotics11060773_

Round 1

Reviewer 1 Report

The manuscript “pruR and PA0065 genes are responsible for decreasing antibiotic tolerance by Autoinducer analog-1 (AIA-1) in Pseudomonas aeruginosa” of Muhammad Reza Pahlevi, Keiji Murakami, Yuka Hiroshima, Akikazu Murakami, and Hideki Fujii proposes that pruR and PA0066-65-64 are members of the antibiotic tolerance suppressors.

No significant comments. I believe it is an interesting manuscript and I recommend that it could be published.

Regards

Author Response

Thank you very much for your excellent comments. It was very encouraging. We have carefully reviewed the manuscripts.

Reviewer 2 Report

The manuscript is well written and organized. The authors investigated the mechanisms of AIA-1 in antibiotic tolerance of Pseudomonas aeruginosa and, given the expertise of the research group in antibiotic tolerance (according to the referenced papers, which should be reduced to the essential self-citations), they were able to identify two suppressor candidates, by which AIA-1 affects antibiotic tolerance.

My main issue is that some aspects could be further discussed, but I understand if the authors do not want to hypothesize without being able to validate such arguments.

Specific comments:

  1. Tn5-PA00065 had a higher effect on survival rate than Tn5-pruR, any hypothesis why? With in silico predictions pointing out the functions of the operon, why should it impact more antibiotic tolerance?
  2. I believe the authors should include statistical analysis even in the first results and include the significant differences in the corresponding Figures (Figures 1-3). I understand we can compare between “conditions” (Biapenem and Biapenem + AIA-1) and also tested strains, but the authors should represent those that are statistically significant given the comparison deemed more relevant for the aim of the study.
  3. In my opinion, Table 1 should also include the SD for the determined concentrations. 
  4. Just to clarify, the construction of the deletion mutant (L. 266) was confirmed by whole-genome sequencing? If so, using which technology/platform?

 Minor comments:

- L. 46, I believe there is no need to have P. aeruginosa between parentheses, as that is the usual abbreviation of that bacterial species.

- Throughout the manuscript, the usage of italics for species and gene/operons should be ensured. Particularly, “Pseudomonas” in L. 181-182, and the genes in L. 315 should be in italic.

- The antibiotic “gentamycin” should be replaced by “gentamicin”.

- CLSI standard M07 has been updated since the 8th edition in 2009. The authors could update reference 36 for the 11th edition (2018).

Author Response

Reviewer 2’s comments

  1. Tn5-PA00065 had a higher effect on survival rate than Tn5-pruR, any hypothesis why? With in silico predictions pointing out the functions of the operon, why should it impact more antibiotic tolerance?

Answer: Our group found that the Tn5-PA0065 had three times higher survival rate than that in Tn5-pruR after exposure to biapenem and AIA-1. We thought that PA0065 and pruR have their independent pathway in antibiotic tolerance. PA0065 (or PA0066-65-64) might had the relation to the major regulator of stress response, rpoS. In this study, we found that ΔPA0066-65-64 exhibited high expression of rpoS compared to PAO1, which might also occur in Tn5-PA0065. Disruption of PA0065 increase the expression of rpoS, thus increasing the antibiotic tolerance of Tn5-PA0065. However, pruR may affect antibiotic tolerance through another pathway.

  1. I believe the authors should include statistical analysis even in the first results and include the significant differences in the corresponding Figures (Figures 1-3). I understand we can compare between “conditions” (Biapenem and Biapenem + AIA-1) and also tested strains, but the authors should represent those that are statistically significant given the comparison deemed more relevant for the aim of the study.

Answer: Thank you for a very good and thoughtful suggestion. In this case, we did not include statistical analysis because our criteria in determining high tolerant mutants was its survival rate ratio to PAO1 after exposure to biapenem+AIA-1. We estimated that strains with at least 10 times higher survival rate than that in PAO1 might exhibited significant differences to PAO1. The problem with statistics was that even lower ratio might resulted in statistically significant result, and this could change our analysis in determining high tolerant mutant. For example, 2 or 3 times higher survival rate than that in PAO1 may be statistically significant, but it does not fulfil our criteria of high tolerant mutant. Thus, we thought that including statistical analysis may confuse the reader.

  1. In my opinion, Table 1 should also include the SD for the determined concentrations.

Answer: Thank you for this suggestion, we did not include SD in the MIC and MBC table because we did the experiment several times (n=4), and then we chose the representative value as MIC or MBC. We used these concentrations in this assay: 256 mg/L, 128 mg/L, 64 mg/L, 32 mg/L, 16 mg/L, 8 mg/L, 4 mg/L, 2 mg/L, 1 mg/L, 0.5 mg/L, 0.25 mg/L, and 0.125 mg/L.

  1. Just to clarify, the construction of the deletion mutant (L. 266) was confirmed by whole-genome sequencing? If so, using which technology/platform?

Answer: The construction of the deletion mutant was confirmed by cycle sequencing. We used BigDyeTM Terminator v3.1 Sequencing Kit. To confirm the correct deletion mutant construction, we detected only 500 base pairs of each upstream and downstream of the gene target that was integrated to the PAO1 genomic DNA.

  1. Minor comments had been adjusted according to Reviewer’s suggestion

Reviewer 3 Report

The study presented in this manuscript is demonstrating the role of pruR and PA0065 gene products in decreasing antibiotic tolerance. The study is interesting since it is focusing on mechanism of autoinducer analog 1, that remained unexplored so far. The experiments are well designed, and data provided seems convincing, showing involvement of above-mentioned gene in suppression of antibiotic tolerance.  However, few points are not clear to me and will appreciate if authors can clarify more clearly in manuscript:

-        The author mentioned that more than 3700 PAO1-Tn5 mutants were screened. Do they mean 3700 clones (including replicates) were screened or 3700 clones with 3700 different integration sites were screened?

-        The author mentioned that PA0065 is involved in decreasing antibiotic tolerance, then why the entire operon was deleted (PA0066065-64)? The appropriate control for direct comparison for integration mutant Tn5-PA0065 would be to delete just PA0065. May be deletion of three genes is causing the effect or other two genes are contributing towards decrease in antibiotic tolerance more than the PA0065.

-        While making these integrant and deletion mutants, the cargo vectors used has selection marker Gentamicin. It is not clear form the M&M section and culture conditions if the plasmid was cured or not? Do the tested strains still possess the selection marker? If yes, then the WT PAO1 that was grown in absence of the gentamicin is not an appropriate control since the presence of gentamicin can affect the survival rate and results cannot be directly compared to the WT PAO1. An ideal control in that case to identify a neutral integration site which is comparable to WT.

-        The authors individually tested the genes (pruR and PA0065) and demonstrated their role in antibiotic tolerance suppression. Is there a reason Why the double deletion mutant for these two genes was not involved in study to investigate the cumulative effect? I believe it will be interesting to include the data for double mutant for these two genes if product from both genes is required or product from one gene is enough.

-        The authors demonstrated that deletion of PA0066-65-64 showed increase in rpoS expression that contribute towards antibiotic tolerance. Will the reverse hold true? Will the Deletion of rpoS increase the expression of PA0065? It is worth checking to correlate the dependence of both genes over each other.

-        Add more information in legends. For example, Fig. 4 mention What is the control.

-        M&M section is not clear and can be improved. For example, it is mentioned that plasmid has are operon, but it is not clear if expression was constitutive or arabinose inducible in culture? It is also not clear in what volume the cultures and all the experiments were performed.  What are centrifuge tubes?

-        Survival rates were quantified 24 h for biapenem and only 4 hours for levofloxacin and tobramycin? Why different time points for quantification? Is 4-hour incubation sufficient?

Round 2

Reviewer 3 Report

The authors have provided response to all points raised by me in my previous review to my satisfaction. Therefore, I endorse the manuscript for publication in its current form.